

# No changes in the abundance of selected fecal bacteria during increased carbohydrates consumption period associated with the racing season in amateur road cyclists

Jakub Wiącek[1], Joanna Szurkowska[1], Jakub Kryściak[2], Miroslawa Galecka[3] and Joanna Karolkiewicz[1]

[1] Department of Food and Nutrition, Poznan University of Physical Education, Poznań, Greater Poland, Poland
[2] Department of Physiology and Biochemistry, Poznan University of Physical Education, Poznań, Greater Poland, Poland
[3] Institute of Microecology in Poznan, Poznań, Greater Poland, Poland

Corresponding author
Jakub Wiącek, wiacek@awf.poznan.pl

## ABSTRACT

**Background**. Cyclists often use high-carbohydrate, low-fiber diets to optimize the glycogen stores and to avoid the gastrointestinal distress during both, the trainings and the competitions. The impact of such dietary changes on gut microbiota is not fully known.

**Methods**. We assessed the abundances of *Faecalibacterium prausnitzii*, *Akkermansia muciniphila*, *Bifidobacterium* spp., and *Bacteroides* spp. and the fecal pH in 14 amateur cyclists during the racing season. Eleven healthy men formed the control group.

**Results**. Despite significant differences in the diet composition and physical endurance levels of amateur cyclists before the competition season (1st term) and control group (carbohydrates: $52.2\% \pm 4.9\%$ *vs* $41.9\% \pm 6.6\%$; VO$_2$max: $56.1 \pm 6.0$ *vs* $39.7 \pm 7.7$; $p < 0.01$; respectively), we did not observe any significant differences in studied gut bacteria abundances or fecal pH between the groups. Although the cyclists' carbohydrates consumption (2nd term) have increased throughout the season ($4.48$ g/kg b.w. $\pm 1.56$ *vs* $5.18$ g/kg b.w. $\pm 1.99$; $p < 0.05$), the studied gut bacteria counts and fecal pH remained unchanged. It seems that the amateur cyclists' diet with increased carbohydrates intake does not alter the gut microbiota, but further research is needed to assess the potential impact of even higher carbohydrates consumption (over 6 g/kg b.w.).

## INTRODUCTION

A long-distance cyclists' training schedule runs year round, but the competitions usually start at the end of March and, due to the required climatic conditions, are continued until October. Exhausting exercise, long-lasting racing season, and difficult weather conditions are the main reasons why cycling can be considered one of the most demanding sports (*Lombardi et al., 2013*).

Exercise or competition of long duration require proper nutrition strategy. Planning water and nutrients intake is an important part of the competition preparation, as prolonged exercise may lead to splanchnic hypoperfusion and gut dysfunction (*Van Wijck et al., 2011*). Long-distance cycling puts the human body under extreme stress, as it causes a shift in the distribution of the blood in the body. It increases abdominal organs ischemia, which is potentially harmful to the intestinal epithelium (through mucosal erosion and ischemic colitis) (*Van Wijck et al., 2012*). Local hypoxia depletes adenosine triphosphate (ATP), resulting in cell death and inflammation (*Colgan et al., 2015*). This, combined with hyperthermia, may lead to tight junction (TJ) proteins breakdown, which in turn increases intestinal permeability and endotoxemia. The exercise-induced endotoxemia intensifies the production and release of pro-inflammatory cytokines (*Lambert, 2008*). Endotoxemia may occur due to the "leakage" of undigested food particles, bacteria and pathogens from the intestinal lumen into the blood circulation (*Camilleri, 2019*). Besides the biochemical consequences of highly demanding training and exhausting competition there is an additional factor negatively affecting the intestinal barrier in cyclists. It is the specific body position, in which the cyclists spend most of the time. They lean forward to get as much aerodynamics as possible and it increases the abdominal pressure causing stomach distress (*De Oliveira, Burini & Jeukendrup, 2014*).

Exercise is a potential tool for the gut microbiota modulation, as it may increase the beneficial bacteria abundance and biodiversity. Evidence suggests that effects of exercise on individual taxa are variable, but usually they lead to an increase in commensal taxa such as *Bifidobacterium, Lactobacilli*, and *Akkermansia* (*Hughes & Holscher, 2021*). However, exercise-induced intestinal barrier impairment is an important risk factor for gut microbiota alterations (*Karl et al., 2018*; *Kulecka et al., 2020*). Gut distress during professional endurance sports participation promotes inflammation and affects recovery through the number of metabolic pathways (*Snipe et al., 2018*). It has been previously shown that dietary modifications required for endurance training and competition may alter the gut microbiota profile, intestinal barrier function, and, in turn, affect the athletes' immune system (*Jang et al., 2019*). Strenuous physical exertion may also alter the circulation of bile acids, a group of mediators in the gut microbiota communication (*Meissner et al., 2011*).

Besides the physical activity levels or antibiotic therapies, dietary factors are generally considered as the main determinants of gut microbiota composition (*David et al., 2014*). Carbohydrate, fibre and protein intake gain the most attention in case of macronutrients impact on microbiota, but proper hydration is another important factor often difficult to control in athletes (*Son et al., 2020*; *Campbell & Wisniewski, 2017*). It is recommended for athletes to consume high amounts of monosaccharides to optimize glycogen storage and sustain blood glucose during exercise. Long-distance runners or cyclists are often instructed to avoid excessive dietary fibre intake, as it may slow down the digestion and cause gastrointestinal disturbances during physical activity (*Rodriguez, DiMarco & Langley, 2009*). On the other hand, low dietary fibre intake may lead to a bowel movements deterioration, thus potentially reduce the abundance of *Akkermansia muciniphila* and *Faecalibacterium prausnitzii*, responsible for short chain fatty acids (SCFAs), such as

propionate and butyrate, production (*Simpson & Campbell, 2015*). It is suggested that these microbiota by-products regulate several biological processes, *e.g.*, the inflammation through monocyte/macrophage and neutrophil suppression (*Bermon et al., 2015*). Gut microbiota modulation might become an important direction of research in the field of athletes' recovery (*Petersen et al., 2017*).

The main aim of the study was to determine the potential effects of repetitive and intense cycling and carbohydrates-enriched diet on the abundance of the targeted gut microbiota in road cyclists at the amateur level. Additionally, we have compared targeted bacteria species abundance of similarly aged people with moderate physical activity level and a balanced diet (control group), to the participants from the study group. The primary outcome measure was the comparison of abundance of selected fecal bacteria between cyclists and healthy controls. Body mass and physical capacity indices, as well as fecal pH were considered as secondary outcomes.

## MATERIALS & METHODS

### Participants

The study group ($n = 14$) consisted of amateur competitive cyclists, aged 20–40 years, members of the sports clubs FTI Sports & Fitness Group and Fogt Bikes Team. Eleven age-matched, healthy, non-smoking, moderately trained males formed the control group. Because of potential differences in gut microbiota responses to changes in the diet and training regimen between males and females (*Batacan et al., 2017*; *Bycura et al., 2021*), only male gender was taken into account in the study. Cyclists were included in the study only if their trainings from over last 2 years were focused on competing (at least five times per week, total of 10 h or more).

Exclusion criteria were as follows: inability to follow the study design and schedule, receiving immunosuppressants, immunomodulators, and other medications that might modify the immune response, suffering an acute infection, using antibiotics, androgenic-anabolic steroids, oral antifungal agents, antiparasitic agents, pre- and/or probiotics, or traveling to tropical countries during the last 4 weeks before the study beginning and also drinking alcohol, smoking tobacco, or taking any chronic medication in the last 6 months before the study beginning.

Before entering the study participants were interviewed on their health status, nutrition and dietary supplementation, as well as medical history. All participants were given full information on planned procedures and have signed the written consent before taking part in the study. The study protocol was written in accordance with the Ethics Guidelines of the Declaration of Helsinki. It was approved by the Local Ethics Committee at the Poznan University of Medical Sciences, reference numbers no.173/16. This study was self-funded. Data collection complied with the Helsinki declaration for biomedical research on human subjects.

The procedures were conducted at the beginning (1st term) of the road cyclists' competition season in April in both groups (cyclists and control) and then after the end of the season in October (2nd term) but only in the group of cyclists.

## Nutrition assessment

Participants were asked to record their food and beverages intake throughout a 3-days span. At the beginning of the study all the subjects were instructed on how to avoid underestimation. Additionally, the cyclists group was instructed on how to maintain their dietary patterns during the competition season. Energy and macronutrient intake was assessed using the NUVERO dietary assessment system (Koszalin, Poland).

## Body composition assessment

Body mass was measured using a certified medical digital beam scale WB-3000 (TANITA Corporation, Tokyo, Japan). Body height was measured using a measurement rod HR-001 (TANITA Corporation, Tokyo, Japan). Body composition was assessed before the first meal of the day using a densitometry device GE Lunar Prodigy Primo Full Densitometer (GE Healthcare Technologies, Boston, MA, USA).

## Maximal oxygen consumption

Maximal oxygen consumption (VO2max) was assessed using a graded exercise test (GXT) with a cycloergometer (Kettler® DX1 Pro; Kettler, Ense, Germany). The test started with a 10-min warm-up, then the load was set to 50W (85–90 rpm) with a programmed increase of 20W per every minute. The test ended for a particular subject when he refused to continue because of exhaustion. The following cardiorespiratory parameters were recorded during the test: minute ventilation, ventilation efficiency (VE), carbon dioxide volume ($VCO_2$), oxygen volume ($VO_2$), respiratory exchange ratio (RER) and heart rate (HR), using Oxycon Mobile portable breath-by-breath gas analyzer (Viasys Healthcare, Hochberg, Germany) and Sport Tester (s610i; Polar, Kempele, Finland). Before each test, the gas sensors and the flowmeter were calibrated according to the manufacturer's instructions. Study participants were asked to avoid exercise as well as alcohol or caffeine consumption at least for a day prior to the tests.

## Stool samples analysis

Collection of stool samples from cyclists (1st and 2nd term) and control group subjects was performed to assess the levels of targeted bacteria and fecal pH. Study participants were informed on the procedures described by the producer of device used for microbiota analysis KyberKompaktPRO (Institute of Microecology, Poznan, Poland) and were asked to bring the samples within 24 h of collection. The samples were frozen and remained frozen until purification of selected bacteria DNA was performed. DNA amplification was carried out using polymerase chain reaction (PCR) according to the method described by QIAGEN (Aarhus, Denmark). QIAamp Fast DNA Stool Mini Kit achieve high sensitivity through the PCR inhibitors removal. The anaerobic bacteria presence in stool samples, such as *Faecalibacterium prausnitzi* of the genus *Faecalibacterium, Akkermansia muciniphila* of the genus *Akkermansia, Bifidobacterium* spp. of the genus *Actinobacteria*, and *Bacteroides* spp. of the genus *Bacteroidetes,* was determined using Real-Time PCR with primers and analyser (ABI 7300) delivered by ThermoFisher Scientific (Boston, MA, USA). The number of sequences' copies amplified by PCR for selected bacteria was converted to the final bacterial count per gram of stool. The limit of detection for the evaluated parameters

was $10^2$ [colony forming units (CFU)/g of feces]. The entire real-time PCR methodology was developed and validated by the Institute of Microecology (Poznan, Poland), the same methods and primers were used in the previous study of *Karolkiewicz et al. (2022)*.

## Statistical analysis

Data are presented as mean and standard deviation (SD) with 95% confidence interval (95% CI) and, where it is relevant (distribution different from normal and/or high data variability), as median and interquartile range (IQR; Q1–Q3). The assumption of normality was verified by Shapiro–Wilk test, whereas homogeneity of variance was verified by Levene's test. Unpaired $t$-test or the Mann–Whitney $U$-test for normally or non-normally distributed continuous variables were used to examine differences between the study groups. The Cochran-Cox test was used as the equivalent of the student's $t$-test for unequal variances, Paired $t$-test or the Wilcoxon rank sum test was used to analyze the statistical significance between variables in before- and after the competition period. A $p$-value of less than 0.05 was regarded as significant. Statistical analysis was performed using the STATISTICA 13.0 software package (StatSoft Inc., Tulsa, OK, USA).

## RESULTS

### Body composition and VO$_2$max in cyclists and control group at baseline (1st term of the study)

At the baseline, participants' somatic parameters didn't differ significantly. The cyclists' training experience significantly influenced their aerobic capacity measured by the maximum value of oxygen uptake ($p < 0.01$; Table 1).

### Nutrients intake in cyclists and control group at baseline (1st term of the study)

Figure 1 shows the proportions of nutrients consumed by the study participants. The dietary survey analysis presented significant differences in protein, carbohydrates, sugar and fat consumption between the examined groups ($p < 0.01$; Table 2). Road cyclists ate more calories from carbohydrates, especially from sugar, and less calories from protein and fat than controls. There was no significant difference in daily energy consumption between control group and cyclists at an amateur level before the competition season. The mean fiber intake in both groups reached the lower range of reference values for daily consumption.

### Participants' stool samples analysis at baseline

The performed analysis showed no statistically significant differences in the examined gut bacteria abundances (expressed as decimal logarithm of CFU per gram of feces) or fecal pH values between the control group and amateur cyclists before the competition season. The fecal abundances of *Bifidobacterium* spp., *Faecalibacterium prausnitzii*, and *Akkermansia muciniphila* were below reference values in both groups (Table 3). The analysis of targeted stool bacteria and fecal pH values in cyclists before and after the competition season did not reveal significant changes (Table 3).

**Table 1 Comparison of descriptive characteristics, body composition and VO$_2$max between cyclists and control group at baseline.**

| | Cyclists ($n = 14$) | | Control group ($n = 11$) | | T-test/ Mann–Whitney ($p$-value) |
|---|---|---|---|---|---|
| | Mean ± SD | Median IQR | Mean ± SD | Median IQR | |
| Age (years) | 32 ± 7.5 | 35.5 13 | 29.1 ± 8.2 | 24 15 | 0.3628 |
| Height [cm] | 183.6 ± 7.7 | 184 9 | 181.2 ± 4.1 | 182 7 | 0.3515 |
| Body mass [kg] | 78.4 ± 10.5 | 75 17.6 | 83.3 ± 13.4 | 76.6 27.4 | 0.2983 |
| Body fat mass [kg] | 10.1 ± 5 | 9.7 6 | 13.5 ± 8,5 | 11.3 16.1 | 0.2260 |
| Fat-free mass [kg] | 68.3 ± 8.5 | 66.7 9.9 | 69.5 ± 6,7 | 70.6 10.8 | 0.6855 |
| Total body water [kg] | 51.3 ± 6.6 | 49.7 7.6 | 50.9 ± 4.7 | 50.1 7.9 | 0.8520 |
| VO$_2$max [ml/kg/min] | 56.1 ± 6.0 | 57.2 9.2 | 39.7 ± 7.7 | 42.9 11.1 | 0,0000[*] |

**Notes.**

SD, standard deviation; IQR, interquartile range; VO$_2$max, maximal oxygen up-take.

*$p < 0.01$.

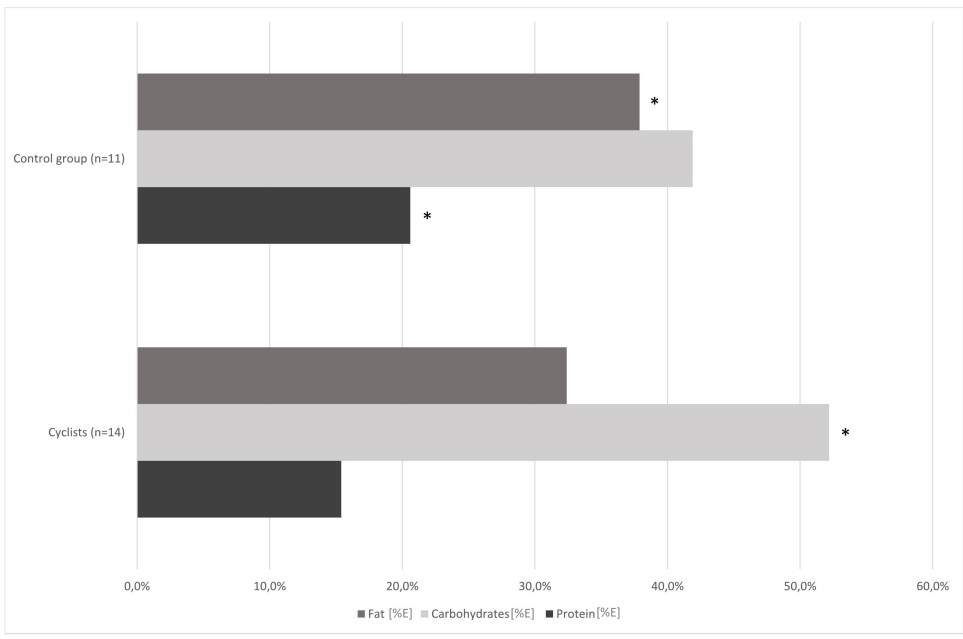

**Figure 1 Comparison of cyclists' and control group diets at baseline.** *—$p < 0.05$.

**Table 2  Comparison of cyclists' and control group daily energy [kcal] and carbohydrates fractions intake.**

| | Cyclists (n = 14) | | Control group (n = 11) | | T-test/ Mann–Whitney (p-value) |
|---|---|---|---|---|---|
| | Mean ± SD | Median IQR | Mean ± SD | Median IQR | |
| Energy [kcal] | 2612.1 ± 873.9 | 2396.3 1 491.9 | 2500.5 ± 1208.0 | 2459.2 1 610.1 | 0.7907 |
| Sugar [%] | 11.1 ± 4.0 | 9.7 4.0 | 5.7 ± 8.0 | 3.7 2.6 | 0.0009[*] |
| Sugar [g] | 69.3 ± 22.7 | 65.2 33.5 | 53.3 ± 110.9 | 25.2 23.8 | 0.0016[*] |
| Fiber [g] | 26.8 ± 13.9 | 24.5 14.9 | 25.6 ± 12.6 | 22.9 23.1 | 0.8481 |
| Lactose [g] | 8.4 ± 8.0 | 6.8 9.1 | 5.6 ± 4.4 | 5.4 7.6 | 0.3961 |
| Starch [g] | 160.6 ± 48.6 | 158.1 57.6 | 133.5 ± 54.6 | 128.3 90.9 | 0.2026 |

**Notes.**
SD, standard deviation; IQR, interquartile range.
[*]$p < 0.01$.

## Cyclists' body composition and physical capacity analysis at 1st and 2nd term (pre- and post-competition season)

After the competition season have finished, significant differences were found in cyclists' total body water content [%]. As shown in Table 4, other body composition characteristics, such as body mass, fat mass or fat-free mass remained unchanged after months of racing. The analysis of differences between the physical capacity measurements in road cyclists pre- and post-competition revealed a significant decrease in cyclists' maximal work load, maximal effort time and VO$_2$max tests (Table 4).

## Cyclists' nutrients intake analysis at 1st and 2nd term (pre- and post-competition season)

The dietary surveys collected from cyclists at the 1st and the 2nd stage of the study revealed significant differences in energy, protein and carbohydrate consumption (Table 5). The further analysis showed that cyclists have covered their increased carbohydrate demands mainly through starchy products intake increase. Small, but insignificant decrease of sugar and fiber intake was observed (Fig. 2).

## DISCUSSION

There is a growing body of evidence indicating that endurance exercise may be an important, external factor that increases gut bacteria species richness (alpha diversity) (*Batacan et al., 2017*). Human studies have reported that subjects with higher cardiorespiratory fitness present higher gut microbiota diversity, especially in case of butyrate-producing bacteria (*Bycura et al., 2021*; *Estaki et al., 2016*; *Barton et al., 2018*; *Allen et al., 2018*). To date, the bacterial production of metabolic mediators, including short-chain fatty acids or secondary

Wiącek et al. (2023), *PeerJ*, DOI 10.7717/peerj.14594

**Table 3** Comparison of targeted stool bacteria and fecal pH values of cyclists and control group, and cyclists at baseline (1st term) and cyclists after the competition season (2nd term).

| | Ref. values | Cyclists 1st term ($n = 14$) | | Cyclists 2nd term ($n = 14$) | | Cyclist 1st term *vs* 2nd term *T*-test/ Wilcoxon (*p*-value) | Control group ($n = 11$) | | Cyclist 1st term *vs* control group *T*-test/ Mann–Whitney (*p*-value) |
|---|---|---|---|---|---|---|---|---|---|
| | | Mean ± SD | Median IQR | Mean ± SD | Median IQR | | Mean ± SD | Median IQR | |
| *Bifidocacterium spp.* | ≥8 CFU/g of feces | 6.9 ± 0.7 | 7.0 1.0 | 7.1 ± 0.5 | 7.0 0.6 | 0.5537 | 6.9 ± 0.7 | 7.0 1.3 | 0.8978 |
| *Bacteroides spp.* | ≥9 CFU/g of feces | 8.8 ± 0.4 | 8.9 0.5 | 8.8 ± 0.4 | 8.9 0.4 | 0.7995 | 8.9 ± 0.6 | 9.0 1.0 | 0.6860 |
| *F. prausnitzii* | ≥9 CFU/g of feces | 8.2 ± 0.5 | 8.3 0.6 | 8.4 ± 0.4 | 8.3 0.2 | 0.2494 | 7.8 ± 0.6 | 7.9 1.0 | 0.0548 |
| *A. muciniphila* | ≥8 CFU/g of feces | 5.6 ± 1.2 | 6.3 0.7 | 6.3 ± 1.3 | 6.5 1.8 | 0.5750 | 6.4 ± 1.1 | 6.9 2.3 | 0.2576 |
| Fecal pH | 6.6 | 6.4 ± 0.6 | 6.5 0.5 | 6.7 ± 0.5 | 7.0 0.5 | 0.1614 | 6.0 ± 0.7 | 6.0 1.0 | 0.0603 |

**Notes.**

SD, standard deviation; IQR, interquartile range; Ref. value, Reference value [Log10 colony forming units (CFU) per gram of feces].

**Table 4  Comparison of descriptive characteristics of body composition and physical capacity (GXT) of cyclists at baseline (1st term) and after the competition season (2nd term).**

| | Cyclists (1st term) | | Cyclists (2nd term) | | T-test/ Wilcoxon (p-value) |
|---|---|---|---|---|---|
| | Mean ± SD | Median IQR | Mean ± SD | Median IQR | |
| Body mass [kg] | 78.4 ± 10.5 | 75 17.6 | 78.5 ± 9.2 | 75.4 18.0 | 0.8329 |
| Body fat mass [kg] | 10.1 ± 5 | 9.7 6.0 | 11.9 ± 4.5 | 10.8 5.2 | 0.1715 |
| Fat-free mass [kg] | 68.3 ± 8.5 | 66.7 9.9 | 67.0 ± 7.0 | 65.4 11.1 | 0.2662 |
| Total body water [kg] | 51.3 ± 6.6 | 49.7 7.6 | 47.5 ± 5.5 | 47.2 6.5 | 0.0721 |
| Total body water [%] | 65.3 ± 7.4 | 63.9 5.6 | 56.1 ± 6.4 | 57.9 8.5 | 0.0019[*] |
| Threshold load [W] | 247.6 ± 49.6 | 250 40 | 250 ± 28.3 | 250 20 | 0.5146 |
| Maximal load [W] | 370 ± 32.3 | 370 40 | 348.6 ± 36.3 | 340 40 | 0.0001[*] |
| Heart rate at threshold [bpm] | 172.1 ± 35.7 | 163.5 13 | 162.6 ± 8.6 | 162 15 | 0.8888 |
| Maximal heart rate [bpm] | 184.4 ± 8.0 | 181.5 8 | 184.2 ± 8.0 | 183 13 | 0.8631 |
| Maximal effort time [min] | 18.9 ± 1.7 | 19 2.3 | 17.8 ± 1.9 | 17.2 2 | 0.0001[*] |
| VO$_2$max [ml/kg/min] | 56.1 ± 6.0 | 57.2 9.2 | 51.4 ± 5.7 | 51.9 8.3 | 0.0001[*] |

Notes.
SD, standard deviation; IQR, interquartile range; bpm, heart beats per minute; VO$_2$max, maximal oxygen uptake.
[*]$p < 0.01$.

**Table 5  Comparison of daily energy [kcal], protein [gkg b.w.], carbohydrates [gkg b.w.], and fats [gkg b.w.] intake of cyclists at baseline (1st term) and after the competition season (2nd term).**

| | Cyclists (1st term) | | Cyclists (2nd term) | | T-test/ Wilcoxon (p-value) |
|---|---|---|---|---|---|
| | Mean ± SD | Median IQR | Mean ± SD | Median IQR | |
| Energy [kcal] | 2612.1 ± 873.9 | 2396.3 1491.9 | 2898.9 ± 900.6 | 2431.4 1258.6 | 0.0231[*] |
| Protein [g/kg b.w.] | 1.32 ± 0.62 | 1.11 0.31 | 1.58 ± 0.50 | 1.43 0.61 | 0.0157[*] |
| Carbohydrates [/kg b.w.] | 4.48 ± 1.56 | 4.34 0.6 | 5.18 ± 1.99 | 4.91 1.70 | 0.0355[*] |
| Fat [g/kg b.w.] | 1.24 ± 0.51 | 1.14 0.45 | 1.39 ± 0.69 | 1.38 0.69 | 0.6934 |

Notes.
SD, standard deviation; IQR, interquartile range.
[*]$p < 0.05$.
b.w., body weight.

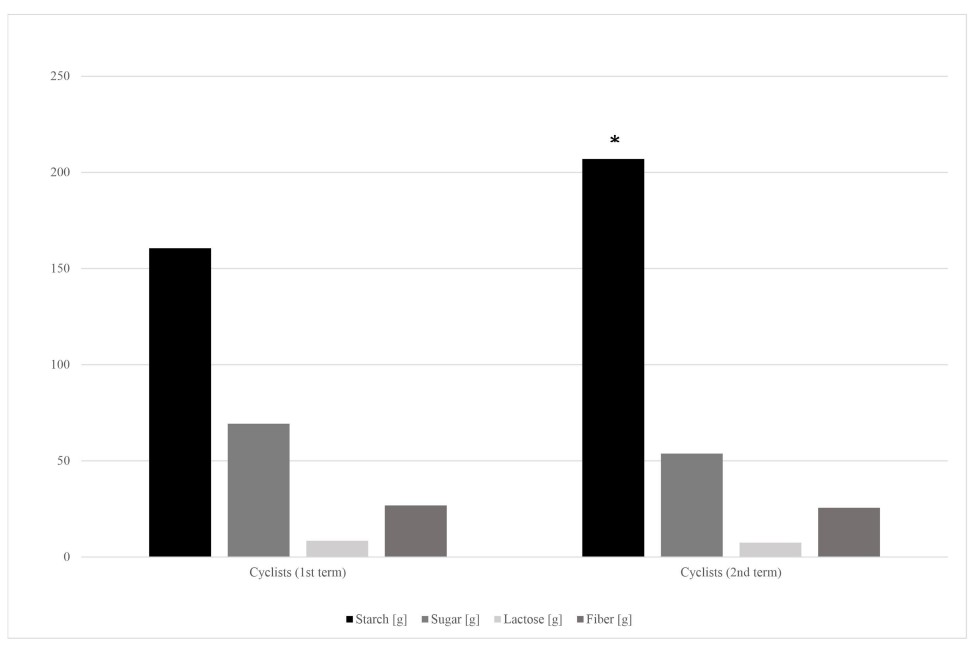

**Figure 2** Carbohydrates' fractions intake in cyclists' diets in the 1st and 2nd term of the study. *—$p <$ 0.05.

bile acids is the most explored mechanism involved in gut microbiota modulation of muscle function (*Frampton et al., 2020*; *Hooper & Gordon, 2001*; *Wahlström et al., 2016*). SCFAs can modulate the glucose metabolism *via* stimulation gluconeogenesis, or through the increase of the glucose, glycogen, and fatty acids bioavailability during exercise (*Ticinesi et al., 2017*). The effects of SCFAs on skeletal muscles metabolism is mediated for example by activating AMP kinase (*Hu et al., 2010*; *Den Besten et al., 2013*).

Although cyclists in our study presented better cardiorespiratory fitness than controls (Table 1), there were no significant differences in terms of targeted gut bacteria abundance (Table 3). A significant decrease in exercise capacity and $VO_2$max was shown in the cyclists after the competition season (Table 4), but it did not result in significant changes of the abundance of bacteria such as *Bifidobacterium* spp., *Bacteroidetes* spp., *Faecalibacterium prausnitzii*, *Akkermansia muciniphila* (Table 3).

It is proposed, that shift in the gut pH may be an important factor in creating the right environment for microbial diversity increase and metabolic functions maintenance. As exercise-induced increased fecal SCFAs concentrations can decrease luminal pH (*Walker et al., 2005*), fecal pH in our study participants was expected to drop. However, there were no significant differences in fecal pH between the group of cyclists and controls, as well as between cyclists in the 1st and the 2nd term of the study (Table 3). Values of fecal pH in study participants remained unchanged and within the neutral range.

It has been previously suggested, that the increase in microbial diversity occurs mostly when a maximal oxygen uptake ($VO_2$ max) is twice as high as $VO_2$ max in sedentary subjects (*Joyner & Coyle, 2008*). In our study the cyclists who competed at an amateur

level had an average pre-competition VO$_2$max of 56.1 ml/kg/min, whereas top-level cyclists' maximum aerobic capacity can reach values as high as 70–85 ml/kg/min (*Faria, Parker & Faria, 2005*). Although the baseline fecal abundances of *Bifidobacterium* spp., *Faecalibacterium prausnitzii,* and *Akkermansia muciniphila* in cyclists were below reference values for these bacteria, the competition season did not affect it significantly (Table 3). In the post-competition tests, we have noticed the effects of accumulated fatigue in athletes, which were proved by a decrease in physical capacity indices in the GXT test in the 2nd term of the study (Table 4).

Besides the endurance training and frequent racing, athletes' nutrition could be an additional factor influencing individual taxa abundances in the gut (*Mohr et al., 2020*). It has been previously found, that diet composition modifies the *A. muciniphila* and *F. prausnitzii*, but there is still more room for improvement in this field of knowledge. Taxa such as *A. muciniphila* and *F. prausnitzii* are proposed to be a potential target in preventing inflammatory and metabolic conditions. However, it's difficult to track the effects of different dietary approaches and particular foods or nutrients on the gut microbiota of athletes, as there are many other factors that could modify the effects (baseline gut microbiota, sex, sport characteristics, energy intake, history of medication and many others) (*Verhoog et al., 2019*).

In our study, the habitual dietary nutrients intake was different in cyclists and control group, and also in cyclists pre- and post-competition season (1st and 2nd term). The cyclists' group (1st term) and control group ate similar amounts of calories, but the cyclists' diet contained significantly more carbohydrates, especially sugar, and less protein and fat (Fig. 1; Table 2).

Energy, protein and carbohydrates intake have significantly increased during the cyclists' racing season (Table 5). Carbohydrate loading is a common strategy used by endurance athletes to maximize glycogen concentrations before a competition (*Burke & Hawley, 2018*). Amateur cyclists' carbohydrates intake fitted the recommended in endurance athletes range, which is 5–12 g/kg b.w. per day (mean 4.48 g $\pm$ 1.56 g/kg b.w. and 5.18 $\pm$ 1.99 g/kg b.w. on the 1st and 2nd term of the study, respectively) (*Jeukendrup, 2011*). It should be noted, that cyclists' in the 1st term of the study almost doubled the control group in terms of saccharose consumption (Table 2). However, it has changed in the competition season, as the cyclists' starch products consumption significantly increased alongside the small and insignificant sugar intake decrease (Fig. 2).

During both, the training and competition seasons, endurance athletes usually limit their intake of indigestible carbohydrates, such as fiber and resistant starch. It is recommended to avoid dietary fiber excess to minimize the risk of potential gastrointestinal problems, such as gas and bloating (*Tiller et al., 2019*; *Rogerson, 2017*). In our study, cyclists' 1st term fiber consumption didn't differ from that of the control group, and from the data taken at the 2nd term of the study (Table 2; Fig. 2). Both groups' mean fiber intake didn't reach the doses recommended in healthy population, which is 30–35 g per day for men (26.8 $\pm$ 13.9 *vs* 25.6 $\pm$ 12.6; cyclists and control group respectively) (*Stephen et al., 2017*). Fiber-consuming, SCFA-producing bacteria abundances, such as *Akkermansia muciniphila*

and *Faecalibacterium prausnitzii*, also were lower than reference values in both groups and both terms of the study (Table 3).

The analysis performed in our study showed that cyclists' daily protein consumption in pre-competition season was $1.3 \pm 0.6$ g/kg b.w., and it has significantly increased to $1.6 \pm 0.5$ g/kg b.w. throughout the competition season ($p < 0.05$; Table 5). This intake levels covered (1st term), and then exceeded (2nd term) the protein consumption recommendations for endurance athletes', which are set within the ranges of 1.2–1.4 (*Vitale & Getzin, 2019*; *Kato et al., 2016*). It should be noted that the protein intake of men in the control group was significantly higher than that of cyclists' and exceeded their RDA (Table 2). The concentration of dietary protein, it's source and amino acid ratio may affect the colonic fermentation processes and furtherly influence the microbial metabolites levels (*Ma et al., 2017*). Despite the significant difference between the groups in terms of protein consumption, there were no statistically significant differences in the examined gut bacteria abundances (Table 3). It stands in line with our previous observations on amateur bodybuilders consuming high-protein diet (*Szurkowska et al., 2021*). Participants of this research have consumed on average 2,1 g of protein per kilogram of body mass (1,58 g in cyclists in the 2nd term), significantly more than sedentary control group, yet had only fecal pH increased.

In presented study, there were significant differences in fat consumption between cyclists and control groups ($p < 0.01$; Table 2). Cyclists' fat intake was $32.4 \pm 5.1\%$ of total energy intake in the pre-competition tests, while during the racing season it was $31.1 \pm 6.2\%$ of total energy, which stands in line with current fat intake recommendations for athletes (approximately 30%) (*Kerksick et al., 2018*). The fat intake of the control group was $37.9 \pm 4.7\%$ of the total energy intake, which significantly exceeded cyclists' fat consumption (Table 2) and the Dietary Reference Intakes (DRIs). High-fat diet may affect the amounts of dietary fat that actually reaches the distal gut, thus affect the gut microbiota (*Rinninella et al., 2019*). An increase in the pool of bile acids that eludes epithelial absorption in the GI tract is one of the mechanisms behind the potential reduction of the abundance of *Faecalibacterium* spp., as they are known as bile sensitive (*Lopez-Siles et al., 2017*; *Mach et al., 2021*). The LCHF (low-carbohydrate high-fat) diet is associated with a significant reduction in the relative abundance of *Faecalibacterium* spp. and an increase in the relative abundance of *Bacteroides* (*Murtaza et al., 2019*). In our study, both groups showed the relative abundances of *Faecalibacterium* spp. below the reference values ($p < 0.01$; Table 3).

## Limitations of the study and future directions

Participants of the study reported the daily diet composition, but the reports did not include the exact methods of preparing food (*e.g.*, cooking or frying, using fresh or frozen products, peeling fruit and vegetables or not, using dairy products with added probiotics or natural, *etc.*). Participants also might not want to honestly report alcohol, nicotine or painkillers, proton-pump inhibitors consumption level. The study design have not addressed the need for "follow-up" measurements over a longer period of time (*e.g.*, next 6 or 12 months). However, mean experience of cyclists was 5,25 years, which is long enough for microbiota to adapt to changes in the diet. The study protocol did not include assessment of bacteria

species and strains different, than *Bacteroides* spp. *Bifidobacterium* spp., *Faecalibacterium prausnitzii*, and *Akkermansia muciniphila.* Other carbohydrate-consuming bacteria levels, such as *Prevotella, Clostriudium* and *Enterococcus* spp., should be investigated in the next experiments assessing the impact of high-carbohydrate diet on athlete's gut microbiota. Future projects should also involve testing the effects of different levels of different nutrients, as well as levels of fiber on cyclists gut microbiota (*e.g.*, comparison with ketogenic or high protein diet, vegan or carnivore).

## CONCLUSIONS

It can be assumed that differences between the groups of amateur cyclists and more sedentary controls in terms of diet composition and physical activity levels were not large enough to distinguish the groups in regard to the gut bacteria composition determined in the study (*Bacteroides* spp. *Bifidobacterium* spp., *Faecalibacterium prausnitzii*, and *Akkermansia muciniphila*). Further research with bigger groups of cyclists with higher or low carbohydrates intake and including follow-up study is needed to clarify the effects of sport-related dietary interventions on gut microbiota composition.

### Funding
This research was funded by a grant "Development of young researchers" obtained from the Poznan University of Physical Education. The funders had no role in study design, data collection and analysis, decision to publish, or preparation of the manuscript.

### Grant Disclosures
The following grant information was disclosed by the authors:
"Development of young researchers" obtained from the Poznan University of Physical Education.

### Competing Interests
The authors declare there are no competing interests.

### Author Contributions
- Jakub Wiącek conceived and designed the experiments, analyzed the data, prepared figures and/or tables, authored or reviewed drafts of the article, and approved the final draft.
- Joanna Szurkowska conceived and designed the experiments, performed the experiments, analyzed the data, authored or reviewed drafts of the article, and approved the final draft.
- Jakub Kryściak analyzed the data, authored or reviewed drafts of the article, and approved the final draft.
- Miroslawa Galecka performed the experiments, authored or reviewed drafts of the article, and approved the final draft.

- Joanna Karolkiewicz conceived and designed the experiments, performed the experiments, prepared figures and/or tables, authored or reviewed drafts of the article, and approved the final draft.

## Human Ethics

The following information was supplied relating to ethical approvals (*i.e.*, approving body and any reference numbers):

The Bioethics Committee at the Medical University in Poznan approved the study (173/16).

## Data Availability

The raw data are available in the Supplemental Files.

## Supplemental Information

Supplemental information for this article can be found online at http://dx.doi.org/10.7717/peerj.14594#supplemental-information.

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
