# Peer review of "No changes in the abundance of selected fecal bacteria during increased carbohydrates consumption period associated with the racing season in amateur road cyclists"

_PeerJ, doi:10.7717/peerj.14594_

## Round 0.1 · original submission · Major Revisions

Although the reviewers agreed that they find merit and pertinence in the current article, some aspects of the methods and results should be more precise.

Reviewer 1 ·

Basic reporting

Overall, this is a clear manuscript addressing a very important issue within a relatively recent and fast growing field of scientific knowledge. In general, the text is well written, and organized in a friendly format to the reader, under the basic academic conventions.

Some minor points can be easily revised (please see bellow):

1 - Line 44 – Please consider starting the sentence with “Eleven” instead of “11”

2 - Line 65 to 72 – The following statements need to be supported with references:
“Long-distance cycling puts the human body under extreme stress, as it causes a shift in the distribution of the blood in the body. It increases abdominal organs ischemia. which is potentially harmful to the intestinal epithelium. Local hypoxia depletes adenosine triphosphate (ATP), resulting in cell death and inflammation. This, combined with hyperthermia, may lead to tight junction (TJ) proteins breakdown, which in turn increases intestinal permeability and endotoxemia. Endotoxemia may occur due to the “leakage” of undigested food particles, bacteria and pathogens from the intestinal lumen into the blood circulation. The exercise-induced endotoxemia intensifies the production and release of pro-inflammatory cytokines.”

3 - Line 66 – Please replace “It increases abdominal organs ischemia. which is potentially” with “It increases abdominal organs ischemia, which is potentially”

4 – Table 1 and table 4 – In the title, please consider the term “descriptive characteristics” instead of “basic characteristics”.

5 - Figure 1 - Please replace % with the symbol %E, since the graphic refers to the percentage of energy intake from each macronutrient.

6 - Table 3 should include the specific units in wich the values of targeted stool bacteria are expressed.

7 - Since Table 3 presents the analysis of targeted stool bacteria values in cyclists before and after the competition season, the corresponding title should be adjusted, as it only mentions “baseline”.

Experimental design

Despite the relatively small sample, the research nature of the study and the technical requirements of the analyses (data) involved makes this work a valuable contribution for the new insights needed to the field, with implications for general health, sports nutrition intervention/optimization of athletic performance, clinical practice and public awareness.

I think it can also help researchers to better guide future study designs and methods, and therefore bring a deeper understanding of the factors that can modulate/influence the human microbiome, and their interplay, in different life contexts.

Validity of the findings

The conclusion drawn by the authors indeed follows what the study results indicate. However, the authors should acknowledge limitations of the study that could have influenced the results observed: an apparent absence of variation in the selected fecal bacteria related to the variable of interest - CHO intake. Furthermore, in the discussion section, the authors should raise more questions (consider additional factors) that can help to explain the results. Even though those factors were not assessed or controled in the study.

Please see the questions below, just as examples that might help to revise the discussion section of the manuscript.

1 - If the period of time elapsed between the 1st term and 2nd term of the study was longer, would the results be different?

2 - Could a change in the intestinal microbiome, induced by an increase in carbohydrate intake, have been found if other species of bacteria had been selected?

3 - Could there have been any lifestyle changes that the participants (or some of the participants) did not report, and that might have influenced the results?

4 - Could the absence of changes in the amount of selected bacteria reflect the combined effect of the simultaneous variation of two factors (increased carbohydrate intake and physical exercise)? If each of the variables were tested separately, could the results be different?

5 - What role might proteins have played? There was also a statistically significant increase in the intake of this nutrient by cyclists between the 1st term and the 2nd term of the study...

6 - Could the relative and apparent stability of the microbiome in the face of rapid environmental changes (lifestyle) reflect a biological adaptation that has an advantage in terms of human health?

Reviewer 2 ·

Basic reporting

Interesting article, which tries to answer questions related to the increased intake of carbohydrates observed in athletes, namely in their microbiota.
The structure of the article follows the rules of the journal, and all materials have been provided.
Figures and tables are intuitive, of good quality and do not appear to have been altered.

Still, and given some questions that arise in reading the article, I think that a limitations section of the article should be added.

Experimental design

The research question is well defined.
The methodology is well described. However, given the results of body composition in cyclists from the first assessment to the second and caloric intake between assessments, it would be pertinent to highlight some of the limitations that may justify these results.

Validity of the findings

Taking into account the characteristics of the sample, regarding nutritional intake, body composition and physiological characteristics, the results obtained open space for further investigations in this area. As the authors point out, the differences between groups and over time were not large enough to draw conclusions.

Additional comments

Also, some of the results are a bit strange:
- In the group of cyclists, there were no changes in body composition from the beginning of the season to the end of it, which is strange.
- Stranger not to have verified alterations in the corporal composition, that is to be accompanied of a caloric ingestion also without alterations.

I think that, as already mentioned, these situations should be highlighted in a limitations section.

---

## Round 0.2 · accepted · Accept

Both reviewers agreed that the article was improved and held the necessary quality for publication.

Reviewer 2 ·

Basic reporting

No comment.

Experimental design

No comment.

Validity of the findings

No comment.

Additional comments

The authors clearly answer my regards about the article.